

# How we see electronic games

Andrew K. Przybylski[1] and Netta Weinstein[2]

[1] Oxford Internet Institute, University of Oxford, Oxford, United Kingdom
[2] School of Psychology, Cardiff University, Cardiff, United Kingdom

## ABSTRACT

Theories regarding the influences of electronic games drive scientific study, popular debate, and public policy. The fractious interchanges among parents, pundits, and scholars hint at the rich phenomenological and psychological dynamics that underlie how people view digital technologies such as games. The current research applied Martin Heidegger's concept of interpretive frameworks (*Heidegger, 1987*) and Robert Zajonc's exposure-attitude hypothesis (*Zajonc, 1968*) to explore how attitudes towards technologies such as electronic games arise. Three studies drew on representative cohorts of American and British adults and evaluated how direct and indirect experiences with games shape how they are seen. Results indicated this approach was fruitful: negative attitudes and beliefs linking games to real-world violence were prominent among those with little direct exposure to electronic gaming contexts, whereas those who played games and reported doing so with their children tended to evaluate gaming more positively. Further findings indicated direct experience tended to inform the accuracy of beliefs about the effects of digital technology, as those who had played were more likely to believe that which is empirically known about game effects. Results are discussed with respect to ongoing debates regarding gaming and broader applications of this approach to understand the psychological dynamics of adapting to technological advances.

*"Anything that is in the world when you're born is normal and ordinary and is just a natural part of the way the world works. Anything that's invented between when you're fifteen and thirty-five is new and exciting and revolutionary and you can probably get a career in it. Anything invented after you're thirty-five is against the natural order of things."*–Douglas Adams (*Adams, 2002*).

## INTRODUCTION

Electronic games are now a dominant entertainment technology (*Lenhart et al., 2015*). In the span of a generation, electronic gaming has transitioned from a niche activity available to those with access to university mainframes to a widely pursued form of leisure accessible on a range of devices ranging from smartphones to virtual reality headsets (*Parkin, 2013*). Half of households in many developed countries now have gaming consoles and it has been estimated hundreds of millions of hours are now invested in gaming each week (*Dutton & Blank, 2015*; *McGonical, 2011*). This explosion of interest has driven discussions in popular, political, and academic circles regarding the impact technologies such as games could be having on individuals and on society more broadly.

Corresponding author
Andrew K. Przybylski,
andy.przybylski@oii.ox.ac.uk

Indeed, many have heralded the potential cognitive, social, and health benefits of game technology (*Baranowski et al., 2008*; *Wellcome Trust, 2013*), whereas others have speculated that they are a contributing cause of mass-shooting tragedies, and advocate for legislation to limit their availability (*Supreme Court of the United States, 2011*; *Bushman, 2013*; *Coperhaver, 2015*; *Dillio, 2014*). Taken as a whole, evidence regarding the effects of electronic games suggests their influence on players—for good or for ill—is small and inconsistent (*Anderson et al., 2010*; *Elson & Ferguson, 2013*; *Etchells et al., 2016*; *Przybylski, 2014a*; *Sherry, 2001*). Though negative effects are estimated to be modest by both critical researchers (e.g., *Elson & Ferguson, 2013*) and advocates for increased regulation (e.g., *Bushman, 2013*), there remain many have who have polarized attitudes regarding the place of electronic games in society (for a review see *Etchells & Chambers, 2014*). The processes undergirding such polarizations, given the absence of evidence for strong positive or negative influences, are not understood.

This gap in our empirical knowledge is an important one because legislative priorities, parenting decisions, and the scientific study of electronic games may be being shaped by attitudinal dynamics we do not yet understand. The aim of the present work is to study views of electronic games using an approach that capitalizes on philosophical and psychological theory. By doing so, the study aims to build empirically grounded insights concerning the conflicting narratives many have with respect to the influences of electronic games.

## Philosophical approaches

In describing the idea of *interpretive frameworks*—the ways by which people acquire, systematize, and act on their knowledge—the philosopher and phenomenologist Martin *Heidegger (1987)* posited that both direct and indirect experience play fundamental roles in guiding attitudes and beliefs. For Heidegger, both first- and second-hand experience shape *internal accuracy*—the degree to which individuals' beliefs are considered internally valid and consistent. Only first-hand experience informs what he termed *empirical* or *provisional accuracy*—the degree to which individuals' views are concordant and consistent with reality. Said differently, for Heidegger, direct and indirect experiences are as likely to undergird highly subjective attitudes about specific topics, persons, or technologies, but interpretive frameworks grounded in first-hand experience tend towards congruency with objective and externally valid, assessments. In this way indirect experience may be internally accurate, that is, contribute to one's own opinion, but only direct experiences can contribute to a view or attitude which is objectively accurate.

## Psychological approaches

Compatible with the phenomenological approach, a long tradition of psychological research has focused on the ways by which experience shapes attitudes and beliefs (*James, 1890*; *Maslow, 1937*). Research in this tradition has demonstrated that familiar experiences and stimuli are preferred to the unfamiliar (*Zajonc, 1968*). This pattern of observations, framed as the *exposure-attitude hypothesis* and later the *mere exposure effect,* proposes that indirect experience, and even more so, direct experience with objects and stimuli

reduces instinctive fear reactions to novel stimuli (*Bornstein, 1989*, *Zajonc, 1968*). Nearly six decades of research conducted in laboratory and real-world contexts suggest that the salutatory effect of experience on attitudes are most accurately thought of as based in the affective (*Harmon-Jones & Allen, 2001*) and perceptual (*Reber, Winkielman & Schwarz, 1998*) correlates of repeated exposure. Said differently, people tend to be more positively and less negatively disposed to stimuli insofar as exposure eases the effort required to perceive and process their features (*Seamon, Brody & Kauff, 1983*) and evokes positive affect (*Harmon-Jones & Allen, 2001*). Indirect and direct experience do indeed shape what Heidegger termed internal accuracy, but their roles in determining the *external validity*, or accuracy, of perspectives on attitudinal objects like technologies are not well understood.

## How gaming technologies are seen

Preliminary research investigating attitudes about electronic games suggests that a synthesis of these approaches may provide a useful conceptual frame for studying how people process and weigh information relevant to gaming. Two studies conducted with representative cohorts suggest that older people, those who grew up before the rise of electronic gaming and therefore have less indirect experience, are less likely to have direct experiences with games and they are more likely to believe games are a contributing cause violence in real-world contexts (*Harris Interactive, 2013*; *Przybylski, 2014b*). This indicates that the degree of exposure to an entertainment technology may, generally speaking, influence the way it is perceived.

Studies with convenience samples suggest that mere exposure, on a generational level, might influence how people see gaming technologies. For example, research by Kneer and colleagues (*2012*) shows that those who grew up in a time when games were common are less likely to believe games cause people to act violently, regardless of whether they themselves play games. Further, there is reason to think that cognitive fluency may have an effect, as people tend to overestimate the influence of games if they consider them in the abstract instead of drawing on concrete experience with specific games (*Ivory & Kalyanaraman, 2009*).

Given that preliminary evidence suggests familiarity and cognitive fluency with electronic games may shape how they are seen, the lens provided by *Heidegger (1987)* and *Zajonc (1968)* may provide a framework for advancing our understanding of attitudes towards digital technologies. Indirect experience through being part of a cohort that discusses and makes visible aspects of gaming might reduce negative views, whereas direct experience may reduce negative views and foster external accuracy of the real impact of games on people. Building on this theorizing, in the current work we explore how these general, cohort-level, and specific, individual-level patterns of experience with games relate to the internal and empirical accuracy of beliefs held about these technologies.

## Present research

Three studies investigated how exposure to electronic games relates to people's views of gaming technology, drawing on population representative cohorts to gain a broad and externally valid perspective. These studies are first to evaluate the extent to which mere

exposure and direct experience may relate to internally consistent views people have about games (Studies 1 & 2), as well as views that are externally consistent, that is, in line with what is empirically known about the influences of electronic games (Study 3).

## STUDY 1

Study 1 was aimed at investigating how indirect (i.e., cohort-linked) and direct (i.e., personal) experiences with electronic games shape one's internally accurate attitudes, using data collected from a nationally representative sample of adults living in the United Kingdom. Hypothesis 1 was based on earlier research and predicted that cohorts with less direct exposure to electronic games—older people and women in particular—would tend to see them negatively (*Harris Interactive, 2013*; *Przybylski, 2014b*). Hypothesis 2 predicted that those having direct experience with games would generally tend to see them more positively. Finally, because direct exposure and personal experience play a central role in both the approaches of Zajonc and Heidegger, we also tested whether direct experience would mediate the relationship between indirect experience and attitudes (Hypothesis 3).

### Method
#### *Participants and measures*
A nationally representative sample of 959 men and 1,019 women ($M_{age} = 46.89$ years, $SD = 16.54$ years) completed measures as part of their participation in the YouGov United Kingdom panel. Socio-demographic information was derived from panel data and the questions detailed below were presented at random in HTML format. The research presented minimal risk, and was granted clearance by the ethics committee of the Oxford Internet Institute at the University of Oxford (CUREC/1A).

All participants polled for the present research were above 18 years of age and members of the YouGov United Kingdom (Studies 1 & 3) or US (Study 2) panels. Panel participants completed a double opt-in process and agreed to the *YouGov (2015b)* and were contacted as part of their ongoing participation in the YouGov Omnibus. In line with the YouGov terms of service (*YouGov, 2015a*), the investigators did not have access to any uniquely identifying participant information. Participants could contact investigators using by way of email contact at YouGov. No inquiries linked to the present studies were received.

*Direct game experience.* Graded exposure to electronic games was assessed using a single self-report item that asked participants: "How often, if at all, do you play video/computer games?" This question was paired with a 6-point Likert-style scale that 1,941 participants used to respond: 1 = "most days," 2 = "once a week," 3 = "once a month," 4 = "several times a year," 5= "once a year," 6 = "never." Scores were reverse coded so that high scores reflected higher levels of direct game experience (Table 1 presents the frequency of these different levels of experience for all three studies).

*Attitudes about games.* Attitudes towards video and computer games were measured with participant responses to eight statements about electronic games. Participants were asked to rate the extent to which they agreed or disagreed with eight statements such as "They

**Table 1** Frequency of different levels of experience with electronic games.

| | Study 1 (UK) | | | Study 2 (US) | | | Study 3 (UK) | | |
| --- | --- | --- | --- | --- | --- | --- | --- | --- | --- |
| | Total | Men | Women | Total | Men | Women | Total | Men | Women |
| Direct game experience | | | | | | | | | |
| Never, % | 46.4 | 40.3 | 52.2 | 30.6 | 30.1 | 31.1 | 45.8 | 39.6 | 51.7 |
| Once a year, % | 6.6 | 6.7 | 6.5 | 7.0 | 7.8 | 6.2 | 3.9 | 4.2 | 3.6 |
| Several times a year, % | 9.8 | 10.5 | 9.1 | 11.0 | 11.1 | 10.9 | 9.2 | 10.8 | 7.6 |
| Once a month, % | 6.2 | 6.2 | 6.2 | 6.3 | 6.1 | 6.4 | 7.8 | 8.8 | 6.8 |
| Once a week, % | 11.8 | 15.5 | 8.3 | 15.7 | 15.2 | 16.2 | 14.6 | 16.9 | 12.5 |
| Most days, % | 19.2 | 20.7 | 17.8 | 29.4 | 29.7 | 29.2 | 18.7 | 19.7 | 17.7 |

Notes.

Percentages reflect adjusted valid proportions of adults at different levels of game engagement as weighted by representativeness across the United Kingdom (Studies 1 and 3) and United States (Study 2).

are a waste of time," and "They are a good form of entertainment." These question were paired with a 5-point Likert-style response scale ranging from: 1 = "strongly disagree," to 5 = "strongly agree." Principle components analysis with a Varimax rotation of the data showed two attitude factors (see Appendix S1): A four-item negative attitudes factor accounting for 32.77% of observed variance, and a four-item positive attitudes factor accounting for 30.64% of variance. Items were averaged to create a positive attitude score ($M = 3.12$, $SD = 0.81$, $\alpha = .78$) and a negative attitude score ($M = 3.17$, $SD = 0.93$, $\alpha = .81$) for each participant.

## Results
### Preliminary analyses
In line with Hypothesis 1, results from zero-order bivariate analyses indicated that younger people tended to report more regular experience with games, $r = -.25$, $p < .001$, and lower negative attitudes, $r = .27$, $p < .001$, as well as higher levels of positive attitudes, $r = -.22$, $p < .001$, towards games. Although results are identical using either method, point-biserial correlations were used in place of independent samples $t$-tests to aid the comparison of preliminary statistics. Results from these correlations indicated women tended to have less gaming experience, $r = -.12$, $p < .001$, and higher negative attitudes, $r = .19$, $p < .001$, as well as less positive attitudes, $r = -.08$, $p = .001$, towards games. The results from these preliminary correlation analyses lend preliminary weight to the exposure-attitude hypothesis (Hypothesis 1) with respect to games (see Table 2).

### Direct effects on gaming attitudes
To test the expectation that those who have direct experience or exposure to games would see these technologies more positively (our Hypothesis 2), a regression model was tested holding variability in participant age and gender constant. Evidence revealed direct experience accounted for independent and significant variability ($\Delta R^2 = .10$) in positive views of gaming over and above variance linked to participant age and gender, $\beta = .32$, $p < .001$. A second regression model found that controlling for participant age and gender, direct game experience was linked to lower negative attitudes, $\beta = -.32$, $p < .001$. This

**Table 2** Bivariate correlations between observed variables in Study 1.

| | 1. | 2. | 3. | 4. |
|---|---|---|---|---|
| 1. Age | – | | | |
| 2. Gender | .02 | – | | |
| 3. Direct game experience | −.25*** | −.12*** | – | |
| 4. Negative attitude about games | .27*** | .19*** | −.38*** | – |
| 5. Positive attitude about games | −.22*** | −.08** | .36*** | −.45*** |

**Notes.**

Zero-order correlation coefficients weighted by representativeness of participants across the United Kingdom general population.

** $p < .01$.

*** $p < .001$.

result supported that hypothesis, indicating both are significant correlates in their own right, which account for incremental variance ($\Delta R^2 = .09$–.10) beyond socio-demographic cohort factors background.

### Indirect effects on gaming attitudes

A series of models evaluated the indirect effects of cohort-level factors on attitudes toward games to test Hypothesis 3 (see Fig. 1). Results from these models using an asymptotic bootstrapping approach with 1,000 resamples (*Preacher & Hayes, 2008*) indicated that participant age and gender had indirect effects on the extent to which they saw games positively by way of personal exposure with gaming contexts (Table 3, top panel). Young people and men tended to have more experience with games and this differential exposure, in turn, related to more positive views of games, a pattern which accounted for between 12.6% and 14.6% of observed variability in positive perceptions of games. Models evaluating the indirect effect of age and gender on negative views of games showed that direct exposure to games mediated the links between participants' backgrounds and attitudes (Table 3, bottom panel). Older participants and women tended to have less experience with games and this lack of experience, in turn, related to more negative views as these indirect associations ($\Delta R^2$), accounting for between 15.7% and 17.2% of variability in negative attitudes towards games.

## Brief discussion

Findings from this study provided support for applying the psychological and philosophical approaches of Zajonc and Heidegger to understanding attitudes towards gaming technologies. However, in this study attitudes were examined broadly as positive and negative evaluations of technology use. Study 2 was designed to test if these findings generalized to a separate sample, and to increase the scope of the research by testing more policy-relevant views on whether games are causally linked to mass-shooting events (e.g., *Bushman, 2013*), and whether laws are needed to restrict access to games (Brown v. EMA 2011). We once again tested the three hypotheses from Study 1: that indirect exposure would link with more positive attitudes (Hypothesis 1), as would direct exposure (Hypothesis 2), and that direct exposure would mediate the effects with indirect exposure (Hypothesis 3).

**Table 3  Study 1 indirect effects analyses examining direct experience as a mediating factor.**

| Dependent variable (DV) | Independent variable (IV) | Mediating variable (MV) | Total effect IV to DV | Direct effect IV to DV | IV to MV | MV to DV | Indirect effect | Boot-strapped bias corrected 95% CI for indirect paths | | Model variance |
|---|---|---|---|---|---|---|---|---|---|---|
| Y | X | Z | c | c′ | a | b | a*b | Lower level | Higher level | $R^2$ |
| Positive | Age | Direct gaming exp. | −0.012 | −0.008 | −0.029 | 0.126 | −0.004 | −0.005 | −0.003 | .146 |
| Attitudes | Gender | Direct gaming exp. | −0.141 | −0.079 | −0.452 | 0.137 | −0.062 | −0.089 | −0.035 | .126 |
| Negative | Age | Direct gaming exp. | 0.015 | 0.011 | −0.029 | −0.151 | 0.004 | 0.003 | 0.006 | .172 |
| Attitudes | Gender | Direct gaming exp. | 0.323 | 0.248 | −0.457 | −0.164 | 0.075 | 0.044 | 0.109 | .157 |

**Notes.**

Coefficients are shown are non-standardized slopes. Models weighted by representativeness of participants across the United Kingdom general population.

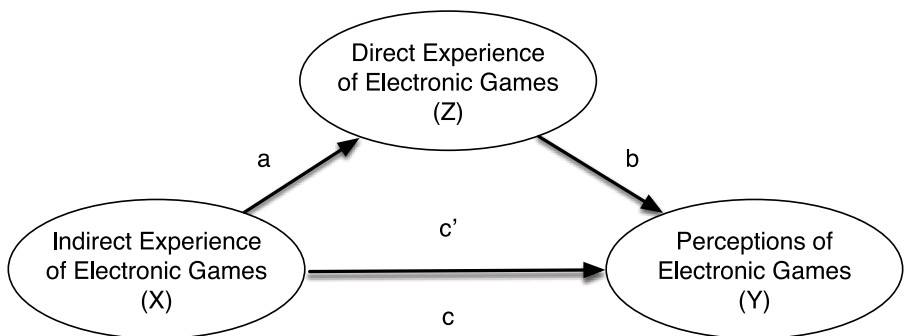

**Figure 1** **Statistical mediation model relations between indirect experience of games and perceptions of games by way of direct experience with games.** Path "a" is the observed effect of the IV to the MV, path "b" is the MV to the DV, path "c" is the total effect of the IV to the DV, and path "c" is the direct effect of the IV to the DV considering variance linked to the MV.

## STUDY 2

### Methods

#### Participants and measures

A nationally representative US sample of 483 men and 517 women ($M_{age} = 48.39$ years, $SD = 16.83$ years) completed the study measures as part of their participation in the YouGov United States panel. Just as was the case in Study 1, socio-demographic cohort-level variables were collected as part of panel participation and measures of direct game experience were unchanged from the first study. A total of 961 participants (96%) responded about their gaming experience.

*Attitudes about games.* Internally valid attitudes towards video and computer games were measured with participant responses to four attitudinal items regarding electronic games (see Appendix S2), similar to Study 1. Principle components analysis showed these four items loaded on a single factor, accounting for 54.48% of observed item variance. Responses to the negatively worded items were reverse coded and averaged with the positive ones to create a single measure of positive attitudes regarding gaming score ($M = 3.29$, $SD = .78$, $\alpha = .70$).

*Games in society.* Two single-item measures were used to assess participants' policy-relevant beliefs concerning electronic games. The first item concerned games and mass violence: "Video/computer games are a contributing cause in mass-shootings," and the second focused on legal regulation of gaming: "New legislation is needed to restrict the availability of video/computer games." Participants used the same 5-point Likert-style scale utilized for general attitude items. A total of 932 participants responded to the item assessing a gaming-shooting link ($M = 2.66$, $SD = 1.26$) and 950 participants responded to the question regarding new gaming legislation with the same 5-point response scale used to assess gaming attitudes ($M = 2.40$, $SD = 1.22$).

**Table 4  Observed variables in Study 2.**

|  | 1. | 2. | 3. | 4 | 5. |
|---|---|---|---|---|---|
| 1. Age | – |  |  |  |  |
| 2. Gender | .03 | – |  |  |  |
| 3. Direct game experience | −.16*** | .01 | – |  |  |
| 4. Attitude about games | −.27*** | −.08* | .40*** | – |  |
| 5. Gaming-Shooting Link | .23*** | .10*** | −.18*** | −.62*** | – |
| 6. New Gaming Legislation | .19*** | .15*** | −.12*** | −.49*** | .64*** |

**Notes.**

Zero-order correlation coefficients weighted by representativeness of participants across the United States general population.

*** $p < .001$.

* $p < .05$.

## Results

### Preliminary analyses

Results from zero-order correlation analyses indicated that socio-demographic background factors related to views of games in line with the exposure-attitude hypothesis (Hypothesis 1; Table 4). Older adults tended to report less regular exposure to games, $r = −.16$, $p < .001$, and less positive attitudes towards games, $r = −.27$, $p < .001$. Older people were more likely to believe that gaming was linked to mass shootings, $r = .23$, $p < .001$, and that laws were needed to restrict games, $r = .19$, $p < .001$. Participant gender showed a similar albeit weaker set of associations indicating that women held less positive attitudes, $r = −.08$, $p = .010$, and were more likely to think games cause shootings, $r = .10$, $p = .002$, and favor laws to limit games, $r = .15$, $p < .001$.

### Direct effects on general attitudes and gaming policy

A hierarchical regression model tested the hypothesis that games would evoke generally positive reactions in those who have had previous experience with them (Hypothesis 2). Positive attitudes about gaming were regressed onto gaming experience, $\beta = .37$, $p < .001$, which accounted for 13.5% ($\Delta R^2$) of variability in this construct over and above participant age, gender, and parenting status. These findings conceptually replicated the findings of Study 1.

Two additional models tested the predictions that personal experience with playing or viewing electronic games is linked to lower likelihood of believing they play a contributing role in mass-shooting or favoring new laws to regulating game availability. Results from these models showed that direct gaming experience was negatively linked to both thinking they contribute to mass-shootings, $\beta = −.15$, $p < .001$, $\Delta R^2 = .02$, and that new laws are needed, $\beta = −.10$, $p = .002$, $\Delta R^2 = .01$, holding variability in participant age and gender constant.

### Indirect effects on general attitudes and gaming policy

A series of analyses examined the indirect effects of age and gender on outcomes of interest using the asymptotic boot-strapping approach employed in Study 1 (Hypothesis 3). Models examining general attitudes about games, beliefs about games and mass-shootings, and thinking new laws are needed to restrict games are presented in Table 5. Results indicated

Przybylski and Weinstein (2016), *PeerJ*, DOI 10.7717/peerj.1931

**Table 5   Study 2 indirect effects analyses examining direct game experience.**

| Dependent variable (DV) | Independent variable (IV) | Mediating variable (MV) | Total effect IV to DV | Direct effect IV to DV | IV to MV | MV to DV | Indirect effect | Boot-strapped bias corrected 95% CI for indirect effect | | Model variance |
|---|---|---|---|---|---|---|---|---|---|---|
| Y | X | Z | c | c′ | a | b | a* b | Lower level | Higher level | $R^2$ |
| Attitude about games | Age | Direct game exp. | −0.012 | −0.010 | −0.017 | 0.133 | −0.002 | −0.003 | −0.001 | .180 |
| | Gender | Direct game exp. | −0.049 | −0.086 | 0.258 | 0.144 | 0.037 | −0.005 | 0.076 | n/a |
| Belief games contribute to shootings | Age | Direct game exp. | 0.018 | 0.016 | −0.019 | −0.076 | 0.001 | 0.001 | 0.003 | .061 |
| | Gender | Direct game exp. | 0.179 | 0.205 | 0.266 | −0.096 | −0.026 | −0.058 | −0.002 | .029 |
| New laws needed to restrict games | Age | Direct game exp. | 0.013 | 0.013 | −0.018 | −0.046 | 0.001 | 0.001 | 0.002 | .036 |
| | Gender | Direct game exp. | 0.349 | 0.365 | 0.248 | −0.063 | −0.016 | −0.041 | 0.001 | n/a |

**Notes.**

All coefficients weighted by representativeness of participants across the general population of the United States.

**Table 6  Observed variables in Study 3.**

|  | 1. | 2. | 3. | 4. | 5. |
|---|---|---|---|---|---|
| 1. Age | – |  |  |  |  |
| 2. Gender | .02 | – |  |  |  |
| 3. Caregiver | −.19*** | .00 | – |  |  |
| 4. Caregiver Co-Play | −.08 | −.12*** | – | – |  |
| 5. Direct game experience | −.23*** | −.10*** | .11*** | .71*** | – |

**Notes.**

Zero-order correlation coefficients weighted by representativeness of participants across the United Kingdom general population.

*** $p < .001$.

that there were significant indirect links for age to general attitudes by way of direct experience ($R^2 = .18$), but no indirect effect was observed for gender (top panel). Direct experience mediated relations linking age ($R^2 = .06$) and gender ($R^2 = .03$) to the belief that video game play contributes to mass-shootings (middle panel). A similar pattern of indirect effects was observed for favoring new restrictive legislation with direct experience mediating links for age ($R^2 = .04$), but no indirect effect was in evidence for participant gender (bottom panel).

## Brief discussion

Findings from this study conceptually replicated and extended the findings from the first study to an American sample and focused on key attitudes driving policy. This is particularly important as it indicates specific, internally held views, on matters of legal policy towards technology may be shaped, in part on the extent to which members of the general public have direct and indirect experience with the technology in question.

The purpose of Study 3 was to go further and investigate the accuracy of perceptions of gaming technology. Although research investigating the influence of games is ongoing (see *Elson & Ferguson, 2013*), researchers largely agree that both the positive and negative effects are small (*Greitemeyer & Mügge, 2014*; *Hull et al., 2014*; *Przybylski, 2014b*), depend on a range of contextual moderators (*Sherry, 2001*), and may not be reliable (*van Ravenzwaaij et al., 2014*). Given these available estimates, this study treated these conclusions from the literature as the ground truth in terms of the Heideggerian concept of empirical accuracy. Following from this, four predictions focused on the empirical accuracy of people's beliefs based on Zajonc's and Heidegger's perspectives were tested.

Hypothesis 1: First, in line with the mere exposure effect and results from Studies 1 and 2 indicating that older participants and women tended to have less direct game experience, it was hypothesized that members of these cohorts would evaluate game effects more negatively and less positively than would be suggested by the scientific literature.

Hypothesis 2: Second, in line with Zajonc and Heidegger, it was predicted that those with direct personal experience of games—in this case, players and caregivers who co-play with their children—would be more likely to hold positive and empirically accurate views. Specifically, it was hypothesized they would be more likely to believe games have small

and inconsistent positive and negative effects—a position well-supported by scientific evidence—because their views are, in aggregate, based in actual experience.

Hypothesis 3: Finally, in line with Heidegger's interpretive frameworks, we hypothesized that first hand experience with games would mediate any links between cohort membership and the empirical accuracy of individuals' evaluations of electronic game effects.

## STUDY 3
### Method
#### *Participants and measures*

A nationally representative UK sample of 929 men and 987 women ($M_{age} = 46.87$, $SD = 16.44$) completed the study measures as part of their participation in the YouGov United Kingdom panel. The measure of direct gaming experience (1,869 responses) was the same as in Studies 1 and 2 (see Table 1). New to Study 3 were measures of caregiver status, caregiver-child co-play, and individual evaluations of positive and negative game effects.

*Caregiver-child co-play.* Participants who were caregivers for a young person aged 18 or younger were asked if they play video/computer games with their child/children. A total of 426 participants (or 22.2%) identified as caregivers, and 211 of these (or 49.5%) reported playing with their child.

*Positive influences of gaming.* Because literatures considering salutatory (*Baranowski et al., 2008*) and negative effects (*Anderson et al., 2010*) are distinct, participants were asked separately about positive and negative influences of gaming. Judgment regarding the potential positive influences of electronic games on young people was assessed with a single item that asked participants to reflect on their beliefs regarding video and computer games. This question was paired with four response options: 1 = "Games have large and significant POSITIVE effects on young people," 2 = "Games have small and inconsistent POSITIVE effects on young people," 3 = "Games have no POSITIVE effects on young people," or 4 = "I don't know." A total of 1,916 participants responded to this item; 13.9% stated they have a large positive effect, 30.1% believed they had no positive effects, and 21.9% said they did not know. Roughly one third of the sample, 34.1%, endorsed an empirically valid response by stating they think games may have small and inconsistent positive effects.

*Negative influences of gaming.* Participants were asked to reflect on their beliefs regarding video and computer games and were provided with four response option: 1 = "Games have large and significant NEGATIVE effects on young people," 2 = "Games have small and inconsistent NEGATIVE effects on young people," 3 = "Games have no NEGATIVE effects on young people," or 4 = "I don't know." A total of 1,916 participants responded to this item; 25.1% stated games have large negative effects, 8.9% believed gaming has no negative effects, and 22.3% said they did not know. Just under half of participants, or 43.7%, provided an empirically accurately estimation by stating they believe games may have small and inconsistent negative effects.

## Results
### Preliminary analyses
Results from zero-order correlation analyses presented in Table 6, indicated that younger adults tended to report more personal experience with games, $r = -.23$ $p < .001$. Women were less likely to co-play games with their children than men, $r = -.12$, $p < .001$, and to report direct experience with games $r = -.10$, $p < .001$. Those who had direct experience with games were also more likely to co-play with their children, $r = .71$, $p < .001$.

### Analytic approach
A series of multinomial logistic regression models examined the effects of indirect and direct experience with interactive gaming technology on holding high, low, and empirically-accurately estimates of the impact of electronic games on young people. This approach minimized the number of statistical tests required and allowed each model to estimate the extent to which different person-level factors would be associated with judgments. Table 7 shows effects for judgments about the nature of positive effects of gaming, and Table 8 shows the judgments concerning the potential negative effects. A series of targeted models evaluated the indirect effects on empirically accurate estimations (Table 9), following the approach used in Studies 1 and 2. These logistic mediation models used a coding of 1 for those who provided an empirically accurate estimate of probable game effects on young people (i.e., small and inconsistent) and 0 for those who provided any other estimate (e.g., large and significant positive effect). These mediation models examined the indirect effects of direct gaming experience on evaluations of gaming technology.

### Direct effects on high estimation of effects
Models examining correlates of estimating large positive effects of gaming indicated co-playing caregivers were slightly more likely ($1.13\times$) than non co-playing parents to think that games have large positive effects on young people. Similar results were found for those who engage games frequently. Compared to individuals without personal experiences playing games, those who played games on a daily ($2.97\times$) and weekly basis ($4.14\times$) tended to be more likely to think games have large positive effects. Multinomial models also showed three groups tended to estimate large negative effects. Compared to men, women were almost twice as likely ($1.79\times$) to report belief in large negative effects of games on young people. Those who played games a few times a year ($2.13\times$) were also more likely to believe that technology has a large negative effect on young people.

### Direct effects on low estimation of effects
Results showed that those who played games several times each year were the only cohort that estimated no positive gaming effects to a significantly greater extent, being 1.75 times more likely to have a lower estimate compared to those who have never played games. Results indicated a number of groups tended to underestimate the potential negative effects of electronic gaming on young people. Men were twice as less likely ($2.18\times$) as women to believe games have no negative effects. Compared to those who never played games, those who played on daily ($2.81\times$), weekly ($2.43\times$), and monthly ($3.56\times$) basis were more

**Table 7  Perceptions of positive game effects.**

| | Empirically accurate evaluation | | | | | | Large effect estimated | | | | | | No effect estimated | | | | | |
|---|---|---|---|---|---|---|---|---|---|---|---|---|---|---|---|---|---|---|
| | *B* | *SE* | LL CI | UL CI | *p* | Exp (B) | *B* | *SE* | LL CI | UL CI | *p* | Exp (B) | *B* | *SE* | LL CI | UL CI | *p* | Exp (B) |
| Gender[a] | | | | | | | | | | | | | | | | | | |
| Men | 0.35 | .126 | 1.11 | 1.81 | .006 | 1.42 | 0.29 | .157 | 0.98 | 1.82 | .066 | 0.99 | −0.20 | .130 | 0.64 | 1.06 | .123 | 1.01 |
| Caregiver[a,b] | | | | | | | | | | | | | | | | | | |
| Not caregiver | −0.34 | .153 | 0.53 | 0.96 | .025 | 0.71 | −0.12 | .195 | 0.60 | 1.30 | .529 | 0.88 | 0.05 | .166 | 0.76 | 01.46 | .743 | 1.06 |
| Caregiver co-play[a,b] | | | | | | | | | | | | | | | | | | |
| No co-play | −1.11 | .282 | 0.19 | 0.57 | <.001 | 0.33 | −1.76 | .381 | 0.08 | 0.35 | <.001 | 0.17 | 0.26 | .319 | 0.70 | 2.44 | .404 | 1.31 |
| Direct game exp.[a,b] | | | | | | | | | | | | | | | | | | |
| Most days | 0.94 | .188 | 1.77 | 3.70 | <.001 | 2.56 | 1.09 | .222 | 1.92 | 4.58 | <.001 | 2.97 | −0.07 | .196 | 0.63 | 1.37 | .709 | 0.93 |
| Once a week | 1.28 | .223 | 2.33 | 5.59 | <.001 | 3.61 | 1.42 | .258 | 2.50 | 6.87 | <.001 | 4.14 | 0.03 | .242 | 0.64 | 1.66 | .899 | 1.03 |
| Once a month | 0.89 | .256 | 1.46 | 3.99 | .001 | 2.41 | 0.56 | .332 | 0.92 | 3.37 | .089 | 1.76 | −0.44 | .295 | 0.36 | 1.15 | .134 | 0.64 |
| Several times a year | 0.56 | .223 | 1.13 | 2.70 | .013 | 1.74 | −0.06 | .323 | 0.50 | 1.77 | .842 | 0.94 | −0.55 | .249 | 0.35 | 0.94 | .026 | 0.57 |
| Once a year | 0.23 | .320 | 0.67 | 2.35 | .477 | 1.26 | −0.15 | .222 | 0.35 | 2.10 | .737 | 0.86 | −0.32 | .325 | 0.38 | 1.37 | .321 | 0.73 |

**Notes.**

[a]Denotes analysis corrects for participant age.

[b]Denotes analysis controls for participant gender. Responses of "I don't know" used as contrast for multinomial logistic models. Coefficients weighted by representativeness of participants across the general population of the United Kingdom.

**Table 8 Perceptions of negative game effects.**

| | Empirically accurate evaluation | | | | | | Large effect estimated | | | | | | No effect estimated | | | | | |
|---|---|---|---|---|---|---|---|---|---|---|---|---|---|---|---|---|---|---|
| | *B* | *SE* | LLCI | ULCI | *p* | Exp (B) | *B* | *SE* | LLCI | ULCI | *p* | Exp (B) | *B* | *SE* | LL CI | UL CI | *p* | Exp (B) |
| Gender[a] | | | | | | | | | | | | | | | | | | |
| Men | 0.05 | .119 | 0.83 | 1.32 | .691 | 1.05 | −0.58 | .136 | 0.43 | 0.73 | <.001 | 0.56 | 0.78 | .192 | 1.50 | 3.18 | <.001 | 2.18 |
| Caregiver[a,b] | | | | | | | | | | | | | | | | | | |
| Not caregiver | −0.33 | .148 | 0.54 | 0.96 | .027 | 0.72 | −0.07 | .177 | 0.66 | 1.32 | .679 | 0.93 | −0.18 | .219 | 0.54 | 1.28 | .402 | 0.83 |
| Caregiver co-play[a,b] | | | | | | | | | | | | | | | | | | |
| No co-play | −1.02 | .277 | 0.21 | 0.62 | <.001 | 0.36 | −0.50 | .326 | 0.32 | 1.15 | .128 | 0.61 | −0.96 | .402 | 0.18 | 0.85 | .017 | 0.38 |
| Direct game exp.[a,b] | | | | | | | | | | | | | | | | | | |
| Most days | 0.94 | .178 | 1.81 | 3.63 | <.001 | 2.56 | −0.17 | .208 | 0.56 | 1.26 | .405 | 0.84 | 1.03 | .263 | 1.68 | 4.70 | <.001 | 2.81 |
| Once a week | 1.20 | .213 | 2.18 | 5.04 | <.001 | 3.32 | 0.39 | .241 | 0.92 | 2.37 | .104 | 1.48 | 0.89 | .309 | 1.33 | 4.46 | .004 | 2.43 |
| Once a month | 0.97 | .268 | 1.56 | 4.47 | <.001 | 2.64 | −0.18 | .339 | 0.43 | 1.63 | .607 | 0.84 | 1.27 | .343 | 1.82 | 6.98 | <.001 | 3.56 |
| Several times a year | 0.39 | .211 | 0.98 | 2.23 | .066 | 1.47 | −0.75 | .281 | 0.27 | 0.81 | .007 | 0.47 | 0.18 | .334 | 0.62 | 2.30 | .591 | 1.20 |
| Once a year | 0.31 | .303 | 0.76 | 2.48 | .302 | 1.37 | −0.15 | .342 | 0.44 | 1.69 | .670 | 0.86 | −0.84 | .730 | 0.10 | 1.81 | .252 | 0.43 |

**Notes.**

[a] Denotes analysis corrects for participant age

[b] Denotes analysis controls for participant gender. Coefficients weighted by representativeness of participants across the general population of the United Kingdom.

**Table 9  Study 3 indirect effects of indirect experience on empirically accurate evaluation of game effects.**

| Dependent variable (DV) | Independent variable (IV) | Mediating variable (MV) | Total effect IV to DV | Direct effect IV to DV | IV to MV | MV to DV | Indirect effect | Boot-strapped bias corrected 95% CI for indirect paths | | Model variance |
|---|---|---|---|---|---|---|---|---|---|---|
| Y | X | Z | $c$ | $c'$ | $a$ | $b$ | $a^a\,b$ | Lower level | Higher level | $^a R^2$ |
| Empirically accurate evaluation of positive influence | Age | Direct game exp. | −0.017 | −0.013 | −0.028 | 0.159 | −0.004 | −0.006 | −0.003 | .051 |
| | Gender | Direct game exp. | −0.112 | −0.054 | −0.336 | 0.176 | −0.059 | −0.102 | −0.028 | .040 |
| | Caregivers | Direct game exp. | 0.429 | 0.353 | 0.519 | 0.171 | 0.223 | 0.051 | 0.141 | .046 |
| | Parent–child co-play | Direct game exp. | 0.796 | 0.496 | 2.604 | 0.117 | 0.305 | −0.089 | 0.699 | n/a |
| Empirically accurate evaluation of negative influence | Age | Direct game exp. | −0.014 | −0.010 | −0.028 | 0.158 | −0.004 | −0.006 | −0.003 | .047 |
| | Gender | Direct game exp. | −0.081 | −0.204 | −0.336 | 0.173 | −0.058 | −0.098 | −0.025 | .040 |
| | Caregivers | Direct game exp. | 0.464 | 0.389 | 0.519 | 0.166 | 0.086 | 0.046 | 0.139 | .047 |
| | Parent–child co-play | Direct game exp. | 0.397 | −0.208 | 2.604 | 0.233 | 0.606 | 0.231 | 1.043 | .046 |

**Notes.**

[a]Coefficients weighted by representativeness of participants across the general population of the United Kingdom.

likely to provide lower estimates of the potential negative effects of games than scientific evidence would suggest.

### Direct effects on empirically-accurate estimation of effects

Importantly, participants varied significantly in terms of their accurate knowledge of gaming effects. Results demonstrated that men were more likely ($1.42\times$) than women to believe games had small and inconsistent positive effects, a belief in line with what is empirically known about their positive effects. This was also the case for caregivers, who were more likely ($1.41\times$) to be accurate compared to their non-caregiving peers. Parents who played with their children were a further three times ($3.03\times$) more likely, compared to non co-play caregivers, to accurately estimate the magnitude of their positive influence. Direct gaming experience also showed strong effects as participants who played games on daily ($2.56\times$), weekly ($3.61\times$), and monthly ($2.41\times$) basis or several times a year ($1.74\times$) were more likely to correctly estimate the actual size of potential positive effects of gaming on young people compared to those who reported no experience with games.

Also importantly, a number of socio-demographic cohorts demonstrated they have a firm handle on the size of the negative effects that electronic games might have on young people. Caregivers in general, and caregivers who play with their children specifically, were more likely to have an accurate idea of their negative effects ($1.39\times$ and $2.78\times$, respectively). Results also showed that personal experience with electronic games was important. Further, those who played electronic games on daily ($2.81\times$), weekly ($2.43\times$), and monthly basis ($3.56\times$), were more likely to correctly estimate the negative effects of gaming compared to those with no direct experience to draw on.

### Indirect effects on empirically-accurate estimation of effects

A series of tests examined direct game experience as a mediator linking age, gender, caregiver status and caregiver-child co-play to reporting empirically-accurate evaluations of the extent to which, for good or ill, gaming contexts shape young people. This approach closely followed the analytic technique used in Studies 1 and 2 to examine gaming attitudes. Though, because the outcome variables were dichotomous (i.e., inaccurate $= 0$ vs. accurate $= 1$), the proportion reduction in error estimates reported are Nagelkerke $R^2$ values in place of adjusted $R^2$.

Indirect effects models (Table 9) indicated that direct gaming experience mediated the links between age, gender, and caregiver status, accounting for approximately 5% of accuracy. Although the direct effect linking caregiver co-play to accuracy regarding the positive effects of gaming was significant, there was no evidence for mediation by way of direct personal experience for these models. Models examining the accuracy of negative gaming effects on young people showed that both forms of experience with gaming contexts served as mediators linking indirect exposure to this outcome. These indirect effects models accounted for roughly 5% of variability in correctly identifying scientifically verifiable negative effects of gaming.

## DISCUSSION

Little is known about the formation of people's attitudes toward gaming technology, yet these attitudes guide the choices made by caregivers, educators, policy-makers, and researchers. Guided by the exposure-attitude hypothesis (*Zajonc, 1968*), the present research synthesized and examined a subset of the promising factors that may guide how people judge the place of games in society. Broadly speaking, the findings derived across three nationally representative samples ($n_{tot} = 4,894$) lent support to the view that perceptions of these entertainment mediums vary systematically across the population as a function of both socio-demographic factors such as age as well as exposure and experience with gaming technologies.

Findings from Studies 1 and 2 suggested that background factors associated with less exposure to electronic games were related to negative attitudes toward games, whereas those able to draw on direct experience with these technologies tended towards skepticism with regards to real-world violence and legislative efforts to restrict the availability of games. Results from mediation analyses indicated both backgrounds and personal experiences were linked to perspectives on gaming across large nationally representative cohorts. This pattern of findings links existing work focused on games to social psychological theorizing on the role of mere-exposure, finding support that those exposed to technology perceive it to be a less negative or potentially harmful influence on individuals, and extends this view in differentiating indirect and direct experiences. Indeed, interpreting these data, it may be that individuals have different opportunities or desires to engage and experience technology as a function of their backgrounds; for example, those who are older were less likely to grow up with easy access to technology.

Findings from Study 3 speak to the challenges faced by parents and those who play electronic games. A number of findings expected by the mere exposure effect were in evidence and speak to the heated debates in the public, academic, and policy arenas. First, most importantly, those we expect might be motivated to ascertain accurate information about the effects of games were also the most likely to correctly estimate their influence on young people. Compared to their non-parent peers, both co-playing and non co-playing caregivers were more accurate in their beliefs regarding gaming effects. A similar pattern was in evidence for adults conversant in the world of gaming; those who were exposed to gaming stimuli on daily, weekly, and monthly bases were the most likely to accurately estimate the magnitude of both the positive and negative effects observed in scientific studies of this technology. Second, as a note of caution, findings suggested some individuals who had direct experience with games were more liable to overestimate the positive and underestimate the possible negative impact of games on young people, whereas the opposite error was observed in those with little gaming experience. These broad tendencies to see games in a biased way—one not grounded in scientific evidence—should be carefully considered by those producing, studying, and crafting regulations concerned with these digital technologies.

These studies speak to how attitudes around technology use are formed, and may guide future work in the area which examines behavior linked to technology use. Social

scientists have long known that attitudes shape behavior, but in sometimes complicated or inconsistent ways (e.g., *Fishbein, 1966*). In future work, researchers could examine how holding the attitudes measured in this study in turn influence which policy decisions are made, and the extent parents and other caregivers support or monitor children's technology use. Indeed, it would be useful to examine how both exposure and pre-existing attitudes toward technology use influence the ways people understand new research findings regarding the effects of technology use on children.

## Research limitations and future directions

Though informative, these studies present two limitations that bear mention. First and foremost, the present studies used cross-sectional data, and as such no causal inferences may be drawn. Indeed, given the nature of the data it is possible that attitudes shape game exposure, not the reverse. Future work tracking engagement with and public opinion regarding interactive technologies, using either a prospective design or selective exposure in a laboratory experiment, both of which would be highly informative. Second, though the pattern of results observed across studies was relatively consistent, the magnitude of these relations were modest, accounting for roughly 4–20% of variance in people's views. It is likely that measures drafted to tap a wider range of gaming experiences sensitive to individuals' history of gaming and preferences would account for more predictive variance in popular attitudes. For example, it would be worthwhile in future research to explore how both quantity and quality (for example, which games people play) of direct experience links to attitudes, and the importance of exposure relative to cultural (e.g., socio-political) and informational (e.g., news sources) factors. Knowing how long people have been playing games, and what kinds of games they have in mind when considering their effects would be greatly informative. As it stands, the present work provides a robust starting point for continued research investigating how cohort- and person-level factors are linked to the attitudes held regarding electronic games.

## Closing remarks

As Douglas Adams observed, technologies are often taken for granted by those who grow up them and are viewed less favorably by those who have little experience with them (*Adams, 2002*). Given that psychological scientists are actively studying how technologies such as social media and electronic games may influence cognition (*van Ravenzwaaij et al., 2014*), emotion (*Przybylski, Rigby & Ryan, 2010*), and aggression (*Elson & Ferguson, 2013*), an empirically grounded understanding of studies and preconceptions about interactive technologies is critical. This work argues and demonstrates that an approach theoretically and empirically grounded in psychological (*Zajonc, 1968*) and philosophical (*Heidegger, 1987*) perspectives provides a fruitful avenue for understanding the place of electronic games in science, society, and policy.

### Funding

The authors received no funding for this work.

## Competing Interests

The authors declare there are no competing interests.

## Author Contributions

- Andrew K. Przybylski conceived and designed the experiments, performed the experiments, analyzed the data, contributed reagents/materials/analysis tools, wrote the paper, prepared figures and/or tables, reviewed drafts of the paper, uploaded data to OSF.
- Netta Weinstein performed the experiments, analyzed the data, wrote the paper, prepared figures and/or tables, reviewed drafts of the paper.

## Human Ethics

The following information was supplied relating to ethical approvals (i.e., approving body and any reference numbers):

The research presented minimal risk, and was granted clearance by the ethics committee of the Oxford Internet Institute at the University of Oxford (CUREC/1A). All participants polled for the present research were above 18 years of age and members of the YouGov United Kingdom (Studies 1 & 3) or US (Study 2) panels. Panel participants completed a double opt-in process and agreed to the *YouGov Terms and Conditions (2015b)* and were contacted as part of their ongoing participation in the YouGov Omnibus. In line with the *YouGov Privacy Policy (2015a)*, the investigators did not have access to any uniquely identifying participant information. Participants could contact investigators using by way of email contact at YouGov. No inquiries linked to the present studies were received.

## Data Availability

Raw data, new variables, and code are available at the Open Science Framework: https://osf.io/kmvs3/?view_only=7f02eae635ac4992a8605db4a1eab5ab.

## Supplemental Information

Supplemental information for this article can be found online at http://dx.doi.org/10.7717/peerj.1931#supplemental-information.

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
