# Peer review of "How we see electronic games"

_PeerJ, doi:10.7717/peerj.1931_

## Round 0.1 · original submission · Minor Revisions

Dear Andrew and Netta,

Thank you for this well constructed and presented piece of work. I'd like to thank the two reviewers for their constructive and clear reviews of the work. Both are positive, requesting some additional points for consideration and clarification, as well as a missing Figure 1. Addressing these comments feels like minor revisions and I look forward to seeing the updated version in the not too distant future!

·

Basic reporting

The research is conducted and reported very effectively and contributes an interesting insight into an ongoing, contentious debate both in the academic literature and public policy and society. The authors write in an academic, but accessible style and the research is presented in a coherent and logical way to aid the reading of the paper.

Experimental design

The research is conducted in a suitable and appropriate manner, with relevant variables included which are conceptually related to the outcome variables of interest. A couple of things which would be useful to clarify or add in to enhance the paper:

1. Perhaps more needed on rationale for H1. That is, what additional evidence is available about why women and older people for example as “less exposed”.

2. As a measure of Direct experience, the number of years experience of gaming would perhaps would be useful too.

Validity of the findings

The validity of the findings are generally sound in relation to the approach of the research. One observation I feel requires further exploration and acknowledgement in the paper however is that the attitudinal measures on games are very general so it is difficult to determine what participants were using as a point of reference when completing these. E.g., some games are excellent at promoting friendship and others aren’t so much. When responding on these items, it is difficult to know what games were being referred to in the responses and thus difficult to determine the consistency of responding across participants. The item on “They develop creativity and thinking skills” may be only relevant for specific types of games, and thus participants’ point of reference for responding is important to note here. With this in mind, to what extent do the authors know that all participants have the same concept in mind in their attitudes about games?

Additional comments

Thank you for your efforts on this research, as this is one area which still remains contentious and this sort of research is greatly needed to dispel many misconceptions surrounding digital gaming effects. I look forward to being able to cite this in my future work!

·

Basic reporting

There are just two (minor) areas that need addressing:

1) The introduction provides a clear and well-written overview of the relevant literature, including good coverage of controversy in the violent video game research area. The sections covering the (a) philosophical and (b) psychological approaches are clearly defined. However, in describing the idea of interpretative frameworks in the philosophical approaches section, the authors concentrate predominantly on the importance of first-hand experience. While this is central to their study, I couldn't help but feel that a very brief overview of how indirect experiences affect attitudes towards things. There is an implicit assumption that readers may have more knowledge about Heideggerian theory than they might actually have, and I think a little bit more context here would be useful.

2) Line 206 refers to Figure 1, but I can't find this anywhere in the manuscript or supplementary materials.

Once these points have been addressed, this article will meet all of the criteria for the 'Basic Reporting' category.

Experimental design

This is a well-designed and rigorous study, which presents a timely and novel investigation into attitudes and beliefs about the use of video games is shaped by exposure to such technology, within the context of both a psychological (mere exposure) and philosophical (direct experience) framework. This is an important relationship to investigate, as there is currently little research that looks at the potential negative impact that a lack of exposure to video games can have on beliefs about their (potentially) deleterious effects. This has important implications for social policy, as the authors point out - but it also has implications for scientific research and media coverage, all of which have the potential to be influenced in quite profoundly negative ways.

As such, the study meets all of the criteria for the Experimental Design category.

Validity of the findings

The authors use a number of analytical approaches, all of which are appropriate, to explore the relationship between video game experience and postive/negative beliefs. Given that this research area is prone to hype and overexaggeration of research findings, the authors should be commended for taking a systematic and objective approach to the analysis, and similarly for taking a measured tone in interpreting the importance of their results. As this is primarily an exploratory/correlational study in nature, the authors rightly note that no causal conclusions should be taken from their findings, and note the present study is a solid starting point for future (and similarly robust) research.

As such, the study meets all of the criteria for the Validity of the Findings category.

---

## Round 0.2 · accepted · Accept

Dear Andrew and Netta,

Thank you for addressing the comments made by the reviewers. I am happy with the accommodation of these points and your reasoning where you have decided not to adjust the work. Considering the minor nature of these adjustments, I haven't felt it necessary for the reviewers to consider these changes and am happy to recommend it for publication in its current form.

I look forward to tracking the impact of the work.